# Neutrophil-to-Lymphocyte Ratio and Platelet-to-Lymphocyte Ratio as Prognostic Markers for Advanced Non-Small-Cell Lung Cancer Treated with Immunotherapy: A Systematic Review and Meta-Analysis

**DOI:** 10.3390/medicina58081069

**Published:** 2022-08-08

**Authors:** Hesti Platini, Eric Ferdinand, Kelvin Kohar, Stephanie Amabella Prayogo, Shakira Amirah, Maria Komariah, Sidik Maulana

**Affiliations:** 1Department of Medical-Surgical Nursing, Faculty of Nursing, Universitas Padjadjaran, Sumedang 45363, Indonesia; 2Faculty of Medicine, Universitas Indonesia, Jakarta 40115, Indonesia; 3Department of Fundamental Nursing, Faculty of Nursing, Universitas Padjadjaran, Sumedang 45363, Indonesia; 4Professional Nurse Program, Faculty of Nursing, Universitas Padjadjaran, Sumedang 45363, Indonesia

**Keywords:** advanced non-small-cell lung cancer, immunotherapy, neutrophil-to-lymphocyte ratio, platelet-to-lymphocyte ratio, prognostic markers

## Abstract

*Background and Objectives*: Advanced non-small-cell lung cancer (NSCLC) has led to a high number of mortalities. Immunotherapy, as a first-line treatment in advanced NSCLC, currently has no clarity regarding its prognostic markers to assess the treatment outcome. This systematic review aimed to evaluate neutrophil-to-lymphocyte ratio (NLR) and platelet-to-lymphocyte ratio (PLR) as prognostic markers in advanced NSCLC patients treated with immunotherapy. *Materials and Methods*: This systematic review was conducted using the PRISMA guidelines, starting from screening for relevant studies from several databases. Each included cohort study was further assessed by using the Newcastle–Ottawa Quality Assessment Scale, and the available data were extracted for qualitative and quantitative synthesis in pooled and subgroup analysis. *Results*: A total of 1719 patients were included in this meta-analysis. Hazard ratio (HR) outcomes for progression-free survival (PFS) and overall survival (OS) for NLR and PLR showed significant results, supporting NLR and PLR as prognostic markers (NLR: HR PFS 2.21 [95% CI: 1.50–3.24; *p* < 0.0001] and HR OS 2.68 [95% CI: 2.24–3.6; *p* < 0.0001]; PLR: HR PFS 1.57 [95% CI: 1.33–1.84; *p* < 0.00001] and HR OS 2.14 [95% CI: 1.72–2.67; *p* < 0.00001]). Subgroup analysis with a cut-off value of 5 for NLR and 200 for PLR also demonstrated notable outcomes. Higher NLR and PLR levels are associated with poor prognostic. *Conclusions*: There is considerable evidence regarding both markers as prognostic markers in NSCLC patients treated with immunotherapy. However, further studies with more homogeneous baseline characteristics are required to confirm these findings.

## 1. Introduction

Lung cancer has become the leading cause of cancer-related mortality. According to the Global Cancer Observatory (GLOBOCAN), in 2020, there were approximately 2.2 million new cases of and 1.8 million deaths caused by lung cancer worldwide [1]. In 2021, the WHO classified lung tumors as epithelial tumors, neuroendocrine neoplasms, and mesenchymal tumors [2]. The most common type of lung cancer is non-small-cell lung cancer (NSCLC), which consists of adenocarcinoma, squamous cell carcinoma, and large cell carcinoma, accounting for 85% of lung cancer. According to the American Joint Committee on Cancer, NSCLC can be divided into four stages; stage I represents the earliest stage, while stage IV represents the advanced and metastases stage [3,4].

Out of all lung cancer patients, 40% of newly diagnosed patients are found in stage IV [5]. Although immunotherapy was reported to be more beneficial than chemotherapy, not all NSCLC cases were effectively treated [6,7]. Hence, prognostic markers are required to assess the immunotherapy outcome in NSCLC patients [8]. Recently, systemic inflammatory markers, such as neutrophil-to-lymphocyte ratio (NLR) and platelet-to-lymphocyte ratio (PLR), were being studied for their usage to evaluate lung cancer prognosis because cancer and tumor development are associated with increased inflammation [9,10]. These markers can be directly obtained from a routine blood test, which is available worldwide and can be applied in limited-resource settings.

Although many studies showed that NLR and PLR were significantly correlated with standard prognostic value in NSCLC treated with immunotherapy [10,11,12,13], a meta-analysis by Tan et al. suggested that PLR was not significantly associated with standard prognostic value [14]. In contrast, other meta-analyses showed that NLR and PLR may be used as potential prognostic factors in NSCLC patients treated with immunotherapy [15,16,17]. Therefore, there is still no clarity regarding NLR and PLR as prognostic markers in NSCLC treated with immunotherapy. Further, there is still an urgent need to find effective and reliable biomarkers to identify patients who most likely benefit from ICI monotherapy rather than chemoimmunotherapy, for example in patients who cannot withstand the chemotherapy. We hypothesized that NLR and PLR are effective to PFS and overall survival OS. Therefore, in this systematic review and meta-analysis, we aimed to evaluate the effectiveness of NLR and PLR to progression-free survival (PFS) and overall survival (OS) in determining the prognosis of advanced NSCLC treated with immunotherapy.

## 2. Materials and Methods

### 2.1. Design Study

The systematic review was conducted and reported using the Preferred Reporting Items for Systematic Review and Meta-Analysis guidelines (see Appendix A) [18].

### 2.2. Study Eligibility Criteria

Screening processes based on eligibility, inclusion, and exclusion criteria were used to filter every research article obtained. The authors included prospective and retrospective cohort studies, randomized controlled trials, and case–control studies measuring NLR or PLR as a prognostic marker in advanced NSCLC treated with immunotherapy. Studies were considered to be excluded if they fulfilled any of the following criteria: (1) treatment with other than immunotherapy, (2) review articles or others; (3) animal studies; (4) inaccessible full-text articles; and (5) non-English articles.

### 2.3. Search Strategy

A literature search was conducted independently throughout several databases, including PubMed, Science Direct, EMBASE, EBSCO, and Cochrane. Any discrepancies were further discussed among the authors. The keywords used in each database were PubMed: (“Carcinoma, Non-Small-Cell Lung” [Mesh]) AND ((neutrophil-to-lymphocyte OR NLR) AND (platelet-to-lymphocyte ratio OR PLR)) EBSCOhost: (“Non-Small-Cell Lung Cancer” OR “NSCLC”) AND (“immunotherapy” OR “Immune Checkpoint Inhibitor” OR “ICI” OR “Atezolizumab” OR “Cemiplimab” OR “Durvalumab” OR “Ipilimumab” OR “Nivolumab” OR “Pembrolizumab” OR “CLTA-4” OR “PD-1” OR “PD-L1”) AND (“Neutrophil-to-lympochyte ratio” OR “PLR”) Science Direct: (Non-Small-Cell Carcinoma OR NSCLC) AND ((neutrophil-tolymphocyte OR NLR) AND (platelet-to-lymphocyte ration OR PLR)) Cochrane: (Non-Small-Cell Carcinoma OR NSCLC) AND ((neutrophil-tolymphocyte OR NLR) AND (platelet-to-lymphocyte ration OR PLR)). Consequently, the results were removed for duplication and screened using the pre-determined eligibility criteria.

### 2.4. Study Selection

All articles were independently reviewed based on a PRISMA flow diagram [19]. A screening process was started through title and abstract and continued with full-text screening of selected studies to exclude studies that met the exclusion criteria if any. All selected studies were finally validated to ensure the eligibility for the next step.

### 2.5. Data Extraction and Quality Assesment

All investigators extracted the available data from the included studies. The following data extracted from each study are shown in Table 1, Table 2 and Table 3. Investigators also obtained the outcomes data rated, which are OS and PFS.

Each study was further assessed for its methodological quality by using the Newcastle–Ottawa Quality Assessment Scale (NOS), which appraised for three quality parameters: study selection, comparability of the population and determination of whether the exposure or outcome includes risk of bias [20]. The studies were considered as high quality if they scored ≥7, moderate if 5/6, and low if ≤5.

### 2.6. Pooled Analysis

Hazard ratio on overall survival and progression-free disease was evaluated on each study and categorized based on the level of NLR and PLR (high or low). All extracted data were analyzed using RevMan ver 5.4.1. Summary data and related 95% confidence interval (CI) were then calculated by meta-analysis pooling on log[hazard ratio] and standard of error (SE). All results were then visualized into forest and funnel plots. The indexes of heterogeneity (X^2^ or Q according to Cochrane, I^2^ and tau^2^) were also calculated to analyze data distribution.

### 2.7. Subgroup Analysis

Due to the variation in cut-off value presented by each study, the investigators further performed a subgroup analysis. The data were divided based on a cut-off value of NLR (5) and PLR (200). Subgroups were analyzed for hazard ratio on overall survival and progression-free disease as the previous pooled analysis.

**Table 1 medicina-58-01069-t001:** Baseline characteristics of included studies.

No	Author; Year	Recruitment Period	Study Characteristics						
			Location	Study Design	Sample Size (Male/Female)	Age Median (Range)/Mean (SD)	Smoking	Types of NSCLC	Types of Immunotherapy
1.	Katayama et al. (2017) [10]	April 2018-November 2019	Japan	Cohort retrospective	81 (44/37)	71 (42–84)	Current/Former = 64 (79%) Never = 64 (79%)	Adenocarcinoma = 50; squamous cell carcinoma = 17; other = 14	Atezolizumab
2.	Suh et al. (2017) [21]	October 2013-April 2016	Seoul	Cohort retrospective	54 (42/12)	Median total = 68 Responder (*n* = 18) = 70 (55–78) Non-responder (*n* = 36) = 62 (43–80)	Current/former = 39 Non = 15	Adenocarcinoma = 31; squamous cell carcinoma = 17; adenosquamous cell carcinoma = 2; NSCLC not specified = 2; pleomorphic carcinoma = 1; large cell carcinoma = 1	Nivolumab (*n* = 31) or Pembrolizumab (*n* = 23)
3.	Khunger et al. (2018) [22]	January 2013-October 2016	United States of America	Cohort retrospective	109 (56/53)	67 (45–90)	Current = 14 (12.8%); Former = 78 (71.6%); Never = 17 (15.6%)	Adenocarcinoma = 71; squamous cell carcinoma = 26; other = 12	Nivolumab
4.	Nakaya et al. (2018) [23]	January 2015-December 2016	Japan	Cohort retrospective	101 (78/23)	69 (45–84)	Current/Former = 84 Never = 16	Squamous = 37; Nnon-squamous = 64	Nivolumab
5.	Russo et al. (2018) [24]	N/A	Italy	Cohort retrospective	28 (25/3)	69 (47–48)	Current/Former = 26	Squamous cell carcinoma = 18; adenocarcinoma = 10	Nivolumab (28) or Docetaxel (34)
6.	Svaton et al. (2018) [25]	2015–2016	Czech Republic	Cohort retrospective	120 (71/49)	≤65 years (*n* = 53) >65 years (*n* = 67)	Current = 55 (45.8%) Former = 43 (35.8%) Non = 22 (18.3%)	Squamous cell carcinoma = 40; adenocarcinoma = 80	Nivolumab
7.	Takeda et al. (2018) [26]	January 2016-October 2017	Japan	Cohort retrospective	30 (19/11)	71 (54–83)	Current = 9 (30%) Former = 17 (56.7%) Never = 4 (13.3%)	Adenocarcinoma = 21; squamous cell carcinoma = 9	Nivolumab
8.	Zer et al. (2018) [27]	May 2013-August 2016	Canada	Cohort retrospective	88 (43/45)	Median (range) = 63.9 (31.1–80.9)	Current = 11 Former = 56 Never = 21	Adenocarcinoma = 66 squamous cell carcinoma = 15 large cell, other = 7	PD-1 axis inhibitors
9.	Pavan et al. (2019) [28]	August 2013-April 2018	Italy	Cohort retrospective	184 (125/59)	67.3 (37.2–83.4)	Current = 160 (87%)	Adenocarcinoma = 108; squamous cell carcinoma = 59; NOS = 14; sarcomatioid = 3	Nivolumab = 145; Pembrolizumab first line = 26, further lines = 6; Atezolizumab = 7
10.	Matsubara et al. (2020) [29]	January 2018-March 2019	Japan	Cohort retrospective	24 (17/7)	64.5 ± 9.7	Current/Former = 17 Never = 7	Adenocarcinoma 18; squamous cell carcinoma = 4; other = 2	Atezolizumab
11.	Prelaj et al. (2020) [30]	August 2015-August 2018	Italy	Cohort retrospective	154 (126/28)	67 (31–86)	Current/former = 128	Squamous = 20; Non-squamous = 6	Nivolumab and Pembrolizumab
12.	Russo et al. (2020) [31]	April 2015-May 2018	Italy	Cohort retrospective	187 (137/50)	67 (34–83)	Current/former = 163	Squamous = 86; non-squamous = 101	Nivolumab
13.	Takada et al. (2020) [32]	January 2016-August 2018	Japan	Cohort retrospective	226 (184/42)	Median (range) = 66 (31–88)	Non = 37 Former = 95 Current = 94	Adenocarcinoma = 146 squamous cell carcinoma = 62 others/unknown = 18	Nivolumab (*n* = 131) or Pembrolizumab (*n* = 95)
14.	Yang et al. (2020) [33]	January 2013-December 2017	China	Cohort retrospective	113 (52/61)	Median age: 50 years	Smoker = 33 Never = 80	Adenocarcinoma = 107 others = 6	Crizotinib
15.	Ksienski et al. (2021) [34]	August 2017-June 2019	Canada	Cohort retrospective	220 (99/121)	70 (62.8–76)	Current = 79 (35.9%) Former = 122 (55.5%) Never = 19 (8.6%)	Squamous = 45; non-squamous = 175	Pembrolizumab

N/A, not available.

**Table 2 medicina-58-01069-t002:** Outcomes of included studies (progression-free survival).

No	Author; Year	NLR Cut-Off Value	PLR Cut-Off Value	PFS		HR PFS	
				NLR	PLR	NLR	PLR
1.	Katayama et al. (2017) [10]	H-NLR > 5; L-NLR ≤ 5	H-PLR > 262; L-NLR ≤ 262	H-NLR = 42 days vs. L-NLR = 86 days [95% CI, *p* < 0.001]	H-PLR = 48.5 days vs. L-PLR = 90 days [95% CI, *p* = 0.033]	HR H-NLR vs. L-NLR = 2.47 [95%, 1.50–4.06, *p* < 0.001]	HR H-PLR vs. L-PLR = 1.67 [95% CI, 1.04–2.68, *p* = 0.035]
2.	Suh et al. (2017) [21]	H-NLR ≥ 5; L-NLR <5	H-PLR ≥ 169; L-PLR < 169	L-NLR = 6.1 months vs. H-NLR = 1.3 months [*p* < 0.001]	N/A	HR H-NLR vs. L-NLR = 23.75/1 [95%, 7.56–74.66, *p* < 0.001]	HR H-PLR vs. L-PLR = 1.80/1 [95%, 0.92–3.52, *p* = 0.085]
3.	Khunger et al. (2018) [22]	H-NLR ≥ 5; L-NLR < 5	N/A	N/A	N/A	N/A	N/A
4.	Nakaya et al. (2018) [23]	H-NLR ≥ 3; L-NLR < 3	N/A	H-NLR = 2.1 months vs. L-NLR = 5.3 months [95% CI, *p* = 0.00528] (2 weeks) H-NLR = 2 months vs. L-NLR = 5.3 months [95% CI, *p* = 0.00515] (4 weeks)	N/A	N/A	N/A
5.	Russo et al. (2018) [24]	H-NLR ≥ 3; L-NLR ≤ 3	H-PLR ≥ 160; L-PLR ≤ 160	L-NLR = 4.0 months vs. H-NLR = 1.0 months [95% CI, *p* = 0.204]	N/A	HR H-NLR vs. L-NLR = 1.05 [95% CI, *p* = 0.0924]	N/A
6.	Svaton et al. (2018) [25]	H-NLR > 3.8; L-NLR ≤ 3.8	H-PLR > 169.1; L-PLR ≤ 169.1	L-NLR = 6.1 months vs. H-NLR = 4.1 months [95% CI, *p* = 0.321]	L-PLR = 6.6 months vs. H-PLR = 3.9 months [95% CI, *p* = 0.108]	1.033 (0.985–1.083) [95% CI, *p* = 0.185]	1.000 (0.999–1.001) [95% CI, *p* = 0.663]
7.	Takeda et al. (2018) [26]	H-NLR ≥ 5; L-NLR < 5	H-PLR > 300; M-PLR = 150–300; L-PLR < 150	L-NLR = 67 days [95% CI, 27–111 days] vs. H-NLR = 109 days [95% CI, 4-NA days] (2 weeks) L-NLR = 95 days [95% CI, 50-NA days] vs. H-NLR = 10 days [95% CI, 6-NA days] (4 weeks)	L-PLR = 66 days [95% CI, 24–111 days] vs. M-PLR = 39 days [95% CI, 24-NA days] vs. H-PLR = 110 days [95% CI, 4-NA days] (2 weeks) L-PLR = 65 days [95% CI, 24-NA days] vs. M-PLR = 12 days [95% CI, 6–136 days] vs. H-PLR = 100.5 days [95% CI, 10-NA days] (4 weeks)	HR L-NLR vs. H-NLR = 4.02 [95% CI, 1.345–12.02] (4 weeks)	N/A
8.	Zer et al. (2018) [27]	H-NLR > 4; L-NLR ≤ 4	H-PLR > 200; L-PLR ≤ 200	N/A	N/A	N/A	N/A
9.	Pavan et al. (2019) [28]	H-NLR ≥ 3; L-NLR ≤ 3	H-PLR ≥ 180; L-PLR ≤ 180	L-NLR = 7.4 months [95% CI, 5.0–9,8 months] vs. H-NLR = 3.1 months [95% CI, 2.2–3.9 months)	L-PLR = 7.3 months [95% CI, 4.4–10.2 months] vs. H-PLR = 2.9 months [95% CI, 1.9–4.0 months]	HR L-NLR vs. H-NLR = 0.557 [95% CI, 0.378–0.820, *p* = 0.003]	HR H-PLR vs. L-PLR = 1.709 [95% CI, 1.178–2.478, *p* = 0.005]
10.	Matsubara et al. (2020) [29]	H-NLR ≥ 5; L-NLR < 5	H-PLR ≥ 150; L-PLR < 150	N/A	N/A	N/A	N/A
11.	Prelaj et al. (2020) [30]	H-NLR ≥ 4; L-NLR ≤ 4	N/A	L-NLR = 4.7 months vs. H-NLR = 2.2 months	N/A	HR H-NLR vs. L-NLR = 2.52 [95% CI, 1.72–3.69, *p* < 0.001]	N/A
12.	Russo et al. (2020) [31]	H-NLR ≥ 5; L-NLR ≤ 5	H-PLR ≥ 200; L-PLR ≤ 200	L-NLR = 7.0 months vs. H-NLR = 4.0 months	L-PLR = 7.0 months vs. H-PLR = 4.0 months	HR L-NLR vs. H-NLR = 0.64 [95% CI, *p* = 0.028]	HR L-PLR vs. H-PLR = 0.67 [95% CI, *p* = 0.05]
13.	Takada et al. (2020) [32]	H-NLR ≥ 6.05; L-NLR <6.05	H-PLR ≥ 245; L-PLR < 245	N/A	N/A	H-NLR/L-NLR = 2.13 (1.55–2.92) *p* < 0.0001	H-PLR/L-PLR = 1.32 (0.99–1.76) *p* = 0.0596
14.	Yang et al. (2020) [33]	H-NLR > 2.4; L-NLR ≤ 2.4	H-PLR > 195; L-PLR ≤ 195	L-NLR = 12.73 months [95% CI, 9.9–15.6 months] vs. H-NLR = 7.73 months [95% CI,6.7–8.7] (*p* = 0.018)	L-PLR = 12.57 months [95% CI, 9.9–15.3] vs. H-PLR = 7 months [95% CI,6.2–7.8] (*p* = 0.002)	HR L-NLR vs. H-NLR = 1.576 [95% CI (1.078–2.304), *p* =0.018]	HR L-PLR vs. H-PLR = 1.862 [95% CI (1.257–2.757), *p* = 0.002]
15.	Ksienski et al. (2021) [34]	H-NLR ≥ 6.4; L-NLR < 6.4	H-PLR ≥ 441.8; L-PLR < 441.8	H-NLR = 3.5 months vs. L-NLR = 10 months [95% CI, *p* < 0.001]	H-PLR = 2.9 months vs. L-PLR = 6.7 months [95% CI, *p* < 0.001]	N/A	N/A

PFS = progression-free survival; H-NLR = high neutrophil-to-lymphocyte ratio; L-NLR = low neutrophil-to-lymphocyte ratio; H-PLR = high platelet-to-lymphocyte ratio; L-PLR = low platelet-to-lymphocyte ratio; HR = hazard ratio; N/A = not available.

**Table 3 medicina-58-01069-t003:** Outcomes of included studies (overall survival).

No	Author; Year	NLR Cut-Off Value	PLR Cut-Off Value	OS		HR OS	
				NLR	PLR	NLR	PLR
1.	Katayama et al. (2017) [10]	H-NLR > 5; L-NLR ≤ 5	H-PLR > 262; L-NLR ≤ 262	H-NLR = 98 days vs. L-NLR = NA [95% CI, *p* < 0.001]	H-NLR = 106 days vs. L-PLR = NA [95% CI, *p* < 0.001]	H-NLR = 98 days vs. L-NLR = NA [95% CI, *p* < 0.001]	H-NLR = 106 days vs. L-PLR = NA [95% CI, *p* < 0.001]
2.	Suh et al. (2017) [21]	H-NLR ≥ 5; L-NLR <5	H-PLR ≥ 169; L-PLR < 169	L-NLR = 14.0 months vs. H-NLR = 2.1 months [*p* < 0.001]	N/A	L-NLR = 14.0 months vs. H-NLR = 2.1 months [*p* < 0.001]	N/A
3.	Khunger et al. (2018) [22]	H-NLR ≥ 5; L-NLR < 5	N/A	H-NLR = 24.2 months [95% CI, 16.1–36.2 months] vs. L-NLR = 29.1 months [95% CI, 16.2–40.9 months], *p* < 0.001	N/A	H-NLR = 24.2 months [95% CI, 16.1–36.2 months] vs. L-NLR = 29.1 months [95% CI, 16.2–40.9 months], *p* < 0.001	N/A
4.	Nakaya et al. (2018) [23]	H-NLR ≥ 3; L-NLR < 3	N/A	N/A	N/A	N/A	N/A
5.	Russo et al. (2018) [24]	H-NLR ≥ 3; L-NLR ≤ 3	H-PLR ≥ 160; L-PLR ≤ 160	L-NLR = 6.0 months vs. H-NLR = 2.0 months [*p* = 0.789]	L-PLR = 10.0 months vs. H-PLR = 6.0 months [*p* = 0.756]	L-NLR = 6.0 months vs. H-NLR = 2.0 months [*p* = 0.789]	L-PLR = 10.0 months vs. H-PLR = 6.0 months [*p* = 0.756]
6.	Svaton et al. (2018) [25]	H-NLR > 3.8; L-NLR ≤ 3.8	H-PLR > 169.1; L-PLR ≤ 169.1	L-NLR = 14.2 months vs. H-NLR = 9.2 months [95% CI, *p* = 0.020]	L-PLR = 14.2 months vs. H-PLR = 9.2 months [95% CI, *p* = 0.014]	L-NLR = 14.2 months vs. H-NLR = 9.2 months [95% CI, *p* = 0.020]	L-PLR = 14.2 months vs. H-PLR = 9.2 months [95% CI, *p* = 0.014]
7.	Takeda et al. (2018) [26]	H-NLR ≥ 5; L-NLR < 5	H-PLR > 300; M-PLR = 150–300; L-PLR < 150	N/A	N/A	N/A	N/A
8.	Zer et al. (2018) [27]	H-NLR > 4; L-NLR ≤ 4	H-PLR > 200; L-PLR ≤ 200	L-NLR = 21.4 months vs. H-NLR = 6.8 months [95% CI, *p* = 0.019]	L-PLR = 14.2 months vs. H-PLR = 9.2 months [*p* = 0.019]	L-NLR = 21.4 months vs. H-NLR = 6.8 months [95% CI, *p* = 0.019]	L-PLR = 14.2 months vs. H-PLR = 9.2 months [*p* = 0.019]
9.	Pavan et al. (2019) [28]	H-NLR ≥ 3; L-NLR ≤ 3	H-PLR ≥ 180; L-PLR ≤ 180	L-NLR = 49.3 months [95% CI, 7.4–91.3 months] vs. H-NLR = 17.3 months [95% CI, 12.1–22.5 months]	L-PLR = 36.4 months [95% CI, 16.4–56.4 months] vs. H-PLR = 14.7 months [95% CI, 9.6–19.7 months]	L-NLR = 49.3 months [95% CI, 7.4–91.3 months] vs. H-NLR = 17.3 months [95% CI, 12.1–22.5 months]	L-PLR = 36.4 months [95% CI, 16.4–56.4 months] vs. H-PLR = 14.7 months [95% CI, 9.6–19.7 months]
10.	Matsubara et al. (2020) [29]	H-NLR ≥ 5; L-NLR < 5	H-PLR ≥ 150; L-PLR < 150	N/A	N/A	N/A	N/A
11.	Prelaj et al. (2020) [30]	H-NLR ≥ 4; L-NLR ≤ 4	N/A	L-NLR = 10.1 months vs. H-NLR = 2.6 months	N/A	L-NLR = 10.1 months vs. H-NLR = 2.6 months	N/A
12.	Russo et al. (2020) [31]	H-NLR ≥ 5; L-NLR ≤ 5	H-PLR ≥ 200; L-PLR ≤ 200	L-NLR = 15.0 months vs. H-NLR = 6.0 months	L-PLR = 15.0 months vs. H-PLR = 11.0 months	L-NLR = 15.0 months vs. H-NLR = 6.0 months	L-PLR = 15.0 months vs. H-PLR = 11.0 months
13.	Takada et al. (2020) [32]	H-NLR ≥ 6.05; L-NLR <6.05	H-PLR ≥ 245; L-PLR < 245	N/A	N/A	N/A	N/A
14.	Yang et al. (2020) [33]	H-NLR > 2.4; L-NLR ≤ 2.4	H-PLR > 195; L-PLR ≤ 195	L-NLR = Not reached vs. H-NLR = 15.77 months [95% CI, 11.4–20.1] (*p* = 0.000)	L-PLR = Not reached vs. H-PLR = 12.00 months [95% CI, 9.0–15.1] (*p* = 0.000)	L-NLR = Not reached vs. H-NLR = 15.77 months [95% CI, 11.4–20.1] (*p* = 0.000)	L-PLR = Not reached vs. H-PLR = 12.00 months [95% CI, 9.0–15.1] (*p* = 0.000)
15.	Ksienski et al. (2021) [34]	H-NLR ≥ 6.4; L-NLR < 6.4	H-PLR ≥ 441.8; L-PLR < 441.8	H-NLR = 5.4 months [95% CI, 3.6–8.4 months] vs. L-NLR = 18.9 months [95% CI, 14.2-NR months]	H-PLR = 4 months [95% CI, 3.1–6.2 months] vs. L-NLR = 15.4 months [95% CI, 11.4–20.4 months]	H-NLR = 5.4 months [95% CI, 3.6–8.4 months] vs. L-NLR = 18.9 months [95% CI, 14.2-NR months]	H-PLR = 4 months [95% CI, 3.1–6.2 months] vs. L-NLR = 15.4 months [95% CI, 11.4–20.4 months]

OS = overall survival; H-NLR = high neutrophil-to-lymphocyte ratio; L-NLR = low neutrophil-to-lymphocyte ratio; H-PLR = high platelet-to-lymphocyte ratio; L-PLR = low platelet-to-lymphocyte ratio; HR = hazard ratio; N/A = not available.

## 3. Results

### 3.1. Search Results

The literature search details are depicted on Figure 1. Through searches, 1691 studies were obtained. After removing 48 duplicates, the authors performed title and abstract screening and 23 articles were retrieved for full-text assessments. We further excluded eight studies due to ineligible outcomes. As a result, 15 studies were included in this systematic review. Among these, 11 studies were further analyzed quantitatively.

### 3.2. Characteristics of Included Study and Critical Appraisal

All included studies were cohort retrospective studies and published between 2017 and 2021. Meanwhile, the provided data were collected between 2013 and 2019. Out of 15 included studies, 7 studies were conducted in Asia [10,21,23,26,29,32,33], 4 in Europe [25,28,30,31], 3 in America [22,27,34], and 1 does not mention the location [24]. A total of 1719 patients with a median age of approximately 60–70 years had advanced NSCLC (mostly adenocarcinoma and squamous cell carcinoma). Studies included also show a higher number of patients who are a current or former smokers. All patients received various PD-1 inhibitor immunotherapy. Risk of bias assessment with NOS reports mostly high-quality research in all studies (see Appendix A).

### 3.3. Study Outcome

The summary of the outcome of each included study is listed in Table 1, Table 2 and Table 3.

#### 3.3.1. PFS and OS of NLR

We analyzed the PFS and OS of patients between low-NLR (L-NLR) and high-NLR (H-NLR) as prognostic markers. Figure 2 and Figure 3 show two forest plots depicting the PFS and OS. Nine studies included in the PFS of NLR give a significant HR of 2.21 towards L-NLR [*p* < 0.0001]. However, a notable heterogeneity is found with an I^2^ value of 91%. (Figure 2)**.** NLR usage in low groups shows a promising performance as a prognostic marker, evident in the OS HR of 2.14 [*p* < 0.00001]. On the other hand, the OS reports homogeneity results by an I^2^ value of 37% (Figure 3). To investigate potential publication bias, we analyze the funnel plot of the study depicted in Figure 4. The first funnel plot (A) depicting the PFS of NLR publication bias. Visually, the plot showed asymmetry with two studies (Suh et al. and Russo et al.) outside the triangular region, indicating a high possibility of publication bias. The second funnel plot (B) depicts the OS of NLR publication bias. This graph demonstrated minor asymmetry with each study placed inside the triangular region, indicating a low possibility of publication bias.

#### 3.3.2. PFS and OS of PLR

The forest plots illustrating the PFS and OS of patients between L-PLR and H-PLR are shown in Figure 5 and Figure 6. A total of six and eight studies were analyzed for PFS HR and OS HR, respectively. The PFS of PLR results in a HR of 1.57 [*p* < 0.00001] towards L-PLR, with homogeneity results of 0% I^2^ value (Figure 5). The pooled HR for OS PLR analysis also supports the PFS results significantly with an HR of 2.14 [*p* < 0.00001] (Figure 6). I^2^ test value, nonetheless; gives a moderate heterogeneity with 37%. The investigation of publication bias was performed using a funnel plot in Figure 7. Both figures (A and B) showed minor asymmetry with all studies inside the triangular area, illustrating a low possibility of bias.

#### 3.3.3. Subgroup Analysis in NLR and PLR Cut-Off

A NLR cut-off value of ≥5 and <5 was determined based on majority of studies and previous reviews [13,21,22,26,29,31,35,36]. Analysis of 4 studies with cut-off value ≥5 and 4 studies with cut-off value <5 to PFS (Figure 8) showed a HR of 2.95 [*p* = 0.0007] and 1.62 [*p* = 0.03], respectively. Another subgroup analysis of 6 studies and 5 studies with the same cut-off value (Figure 9) resulted in a HR of 2.97 [*p* < 0.00001] and 3.31 [*p* < 0.00001], respectively.

A PLR cut-off value of >200 and ≤200 was determined from the median of PLR cut-off values extracted from the included studies. A total of 2 studies with a PLR cut-off value >200 and 4 studies with a PLR cut-off value ≤200 were analyzed (Figure 10) and showed a HR of 1.41 [*p* = 0.007] and 1.70 [*p* < 0.00001], respectively. Another subgroup analysis was performed in 2 studies and 4 studies with the same cut-off value (Figure 11), which resulted in 1.41 [*p* = 0.007] and 1.70 [*p* < 0.00001], respectively.

## 4. Discussion

NSCLC is the most common subtype, and most patients usually present as stage IV [15]. The programmed cell death (PD-1) and PD-L1 inhibitors are used as the first-line treatments for advanced or metastatic NSCLC [37]. A prognostic factor is a predictor of the natural history of a disease. Some potential biomarkers are able to predict the prognosis of the disease, such as biometric data and clinical findings, blood parameters, immunohistochemical, and DNA markers. Gender, low-performance status, steroid, antibiotics usage, BMI < 25, and pleural or liver metastases are clinical findings indicating worse prognostic factors. Laboratory values (CRP, enzyme, and blood count) and tumor genetics (p53 and PD-L1) are also useful [6].

Hematological parameters are the most common and easy to obtain in clinical practice [6]. Some recent studies reported NLR and PLR reflect host immune reaction and inflammation process, and are therefore strongly associated with poor clinical outcomes in NSCLC treated with immunotherapy [38].

A study by Petrova et al. explained that neutrophils and platelets had an important role in progression and development of tumor cells through various chemokines and cytokines secretion, such as IL-6, IL-8, TGF, VEGF, and MMP [39]. All would affect directly on tumor cells or indirectly on its microenvironment. In addition, neutrophil acted itself as an inflammatory response, which caused the inhibition of antitumor immune response through CD8+ T cells cytotoxic inhibition. Therefore, laboratory findings will present with neutrophil increase and lymphocyte decrease, which are associated with poor prognosis [38,39].

Erpenbeck et al. reported that platelets were capable of promoting the growth, development, and enlargement of tumors via non-inflammatory mechanisms and angiogenesis, such as stimulation of MMP9, adhesive molecules, and growth factor synthesis [40]. Further, He et al. suggested that platelet might protect tumor cells from immune surveillance (CD8+ T cells) and help them to attach through the endothelium at metastatic sites [41]. Accordingly, a higher PLR value was associated with poor prognostic value.

Our study showed a significant correlation of H-NLR and H-PLR to shorter PFS and OS which is supported by most other meta-analyses. Xu et al. also showed similar results for PLR to PFS (HR = 1.52; *p* < 0.00001) and OS (HR = 1.91; *p* < 0.0001) [12]. Wang et al. also reported NLR to PFS (HR = 1.74; 95% CI: 1.27–1.84) and OS (HR = 2.50; 95% CI: 1.60–3.89) [42]. Tan et al. also reported similar result for H-NLR to shorter PFS and OS; however, no significant result for H-PLR to PFS and OS compared to L-PLR before the initiation of immunotherapy in several types of malignancies [14].

Most of the studies included in this review exhibited distinct cut-off values. Moreover, several factors are known to influence NLR and PLR, including age, gender, ethnic, environmental factors and lifestyle [34,43,44]. A NLR cut-off value of 5 was most commonly used in our reviewed studies [13,21,22,26,29,31]. It was also the most suitable number in western countries and recommended to be applied in daily practice [43]. Our study suggests that a NLR cut-off value of 5 is correlated with a lower PFS and OS than below 5, which is supported by Mei et al. [35]. Although the cut-off standard is still unknown, it is relatively acceptable to use a NLR cut-off value of 5 in general for prognostic determination.

Various PLR cut-off values ranging from 150 to 262 were extracted from the included studies in this review. Our study showed that a PLR cut-off value of 200 is significantly associated with PFS and OS. A similar finding by Gu et al. suggested that a PLR cut-off value of 180 showed significant association with PFS and OS in Caucasians with NSCLC; however, no association was found with a such cut-off value in Asians [43]. Nonetheless, we suggested that a PLR value higher than 200 is correlated with a lower prognostic value. This finding corresponded to Xu et al. that indicated a PLR cut-off value of 170 was associated with decreased prognostic value for PFS [44]. To our knowledge, no study was available to assess PLR to PFS and OS specifically in NSCLC patients treated with immunotherapy.

Prognostic judgments are important to give information to patients about their future and potential outcomes so they can make decisions, either for their health, or others. If proven, these markers may also serve as clinical judgement for treatment failure, associated with worse prognosis. Therefore, it will provide a basis for medical professionals in making rational medical decisions, such as treatment change decision. Proposing NLR and PLR as novel prognostic markers for NSCLC patients treated with immunotherapy, this systematic review and meta-analysis can serve as a reference for prognostic guidelines in the future.

There were some limitations in this study. First, we included studies with various cut-off values, causing a potential bias. Other factors from patients’ baseline characteristics, such as age, gender, and location of study, may also affect the results, shown by significant heterogeneity in a HR of NLR PFS outcomes. Despite the diversity of baseline characteristics, the studies included were retrospective studies, which are the best study design for prognostic analysis. Second, we do not use a failsafe to assess the impact of publication bias due to software limitations. Furthermore, the low risk of bias from the included studies is another strength of our research.

## 5. Conclusions

To summarize, NLR and PLR are promising prognostic markers in NSCLC patients treated with immunotherapy, evident in remarkable PFS and OS hazard ratio outcomes. As stated in our limitations, no standardized cut-off value is available for NLR and PLR. Thus, we suggest making an identical cut-off value for worldwide usage, so more studies conducted with the same value can be made, resulting in a more accurate review in the future. We also recommend creating an updated systematic review and meta-analysis, using more homogeneous baseline characteristics and strengthening our current findings, including proving the definitive correlation of PLR as a prognostic marker.

## Figures and Tables

**Figure 1 medicina-58-01069-f001:**
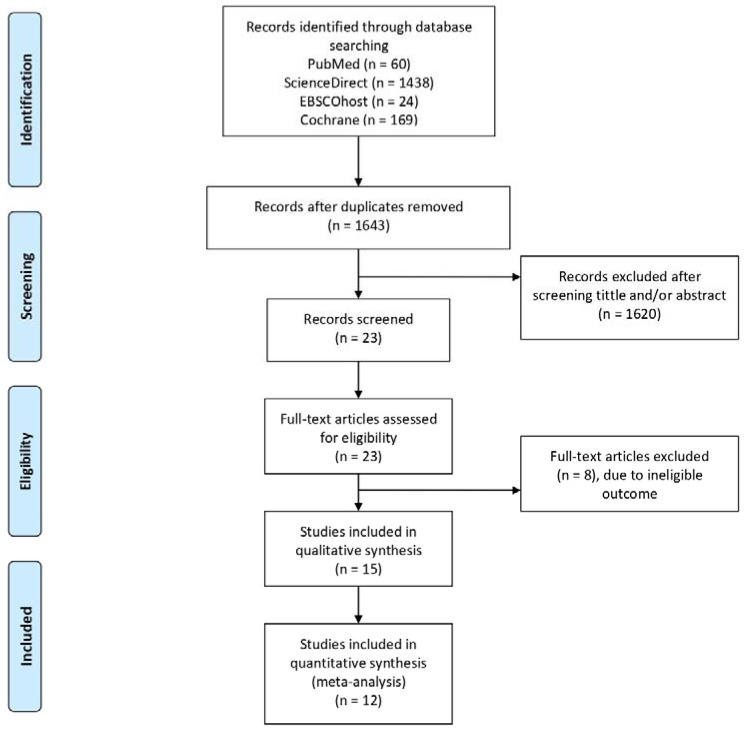
PRISMA flow diagram.

**Figure 2 medicina-58-01069-f002:**
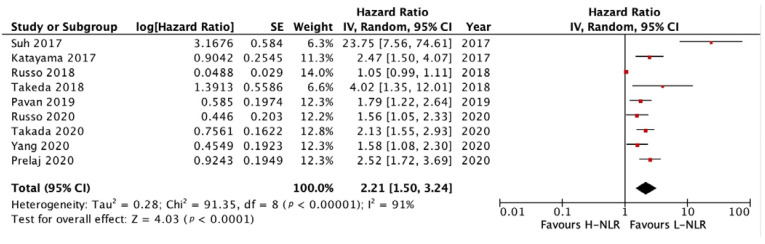
Forest plot H-NLR versus L-NLR to PFS in patients treated with immunotherapy. Red dots represent study weights; the bivalve represent the overall effect.

**Figure 3 medicina-58-01069-f003:**
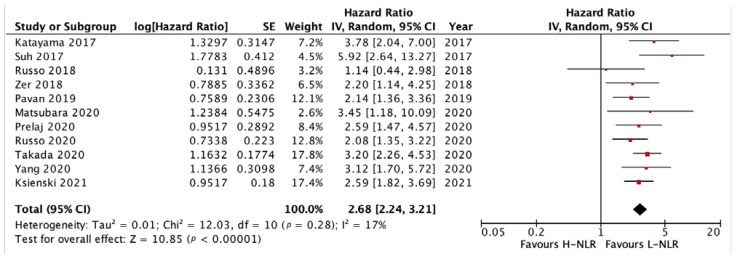
Forest plot H-NLR versus L-NLR to OS in patients treated with immunotherapy. Red dots represent study weights; the bivalve represent the overall effect.

**Figure 4 medicina-58-01069-f004:**
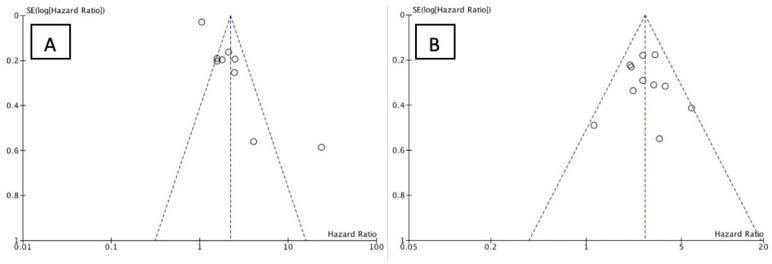
Funnel plot of NLR (**A**) PFS and (**B**) OS. The small circle represent individual study.

**Figure 5 medicina-58-01069-f005:**
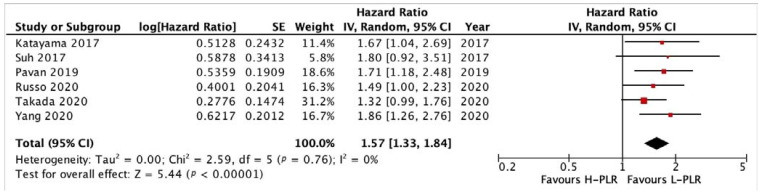
Forest plot H-PLR versus L-PLR to PFS in patients treated with immunotherapy. Red dots represent study weights; the bivalve represent the overall effect.

**Figure 6 medicina-58-01069-f006:**
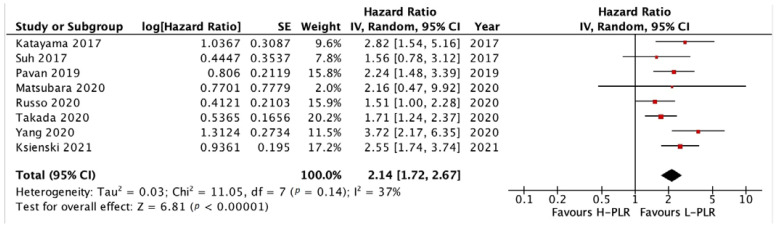
Forest plot H-PLR versus L-PLR to PFS in patients treated with immunotherapy. Red dots represent study weights; the bivalve represent the overall effect.

**Figure 7 medicina-58-01069-f007:**
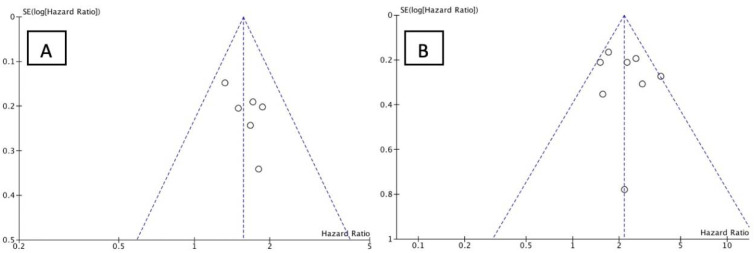
Funnel plot of PLR (**A**) PFS and (**B**) OS. The small circle represent individual study.

**Figure 8 medicina-58-01069-f008:**
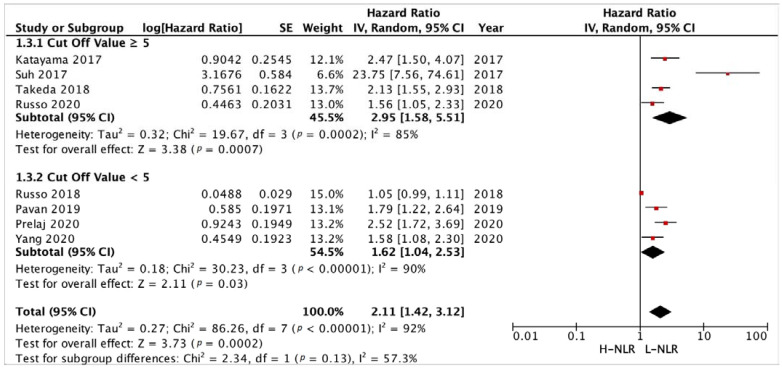
Forest plot cut-off value subgroup analysis of H-NLR versus L-NLR to PFS in patients treated with immunotherapy. Red dots represent study weights; the bivalve represent the overall effect.

**Figure 9 medicina-58-01069-f009:**
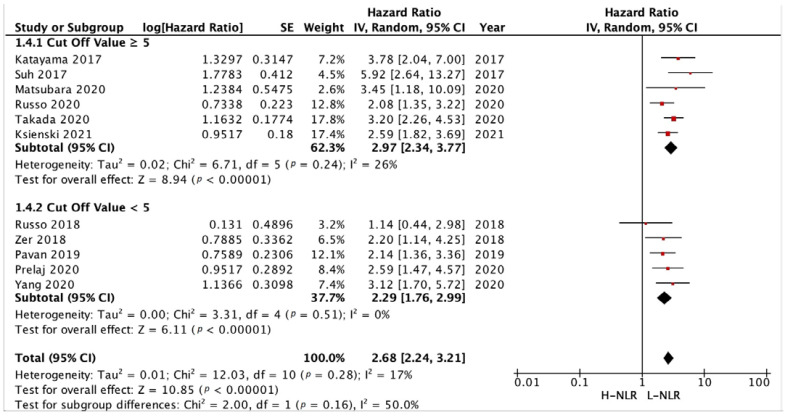
Forest plot cut-off value subgroup analysis of H-NLR versus L-NLR to OS in patients treated with immunotherapy. Red dots represent study weights; the bivalve represent the overall effect.

**Figure 10 medicina-58-01069-f010:**
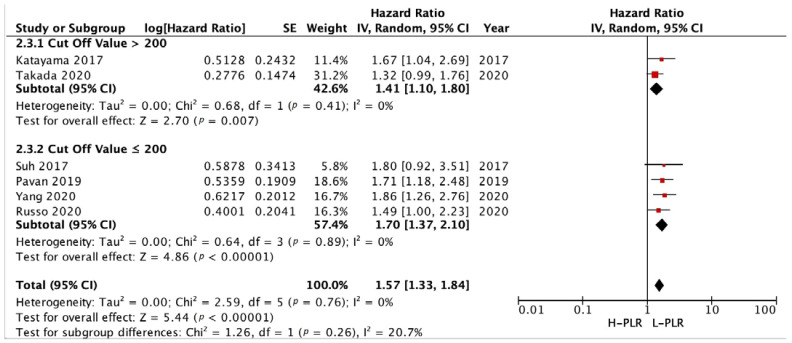
Forest plot cut-off value subgroup analysis of H-PLR versus L-PLR to PFS in patients treated with immunotherapy. Red dots represent study weights; the bivalve represent the overall effect.

**Figure 11 medicina-58-01069-f011:**
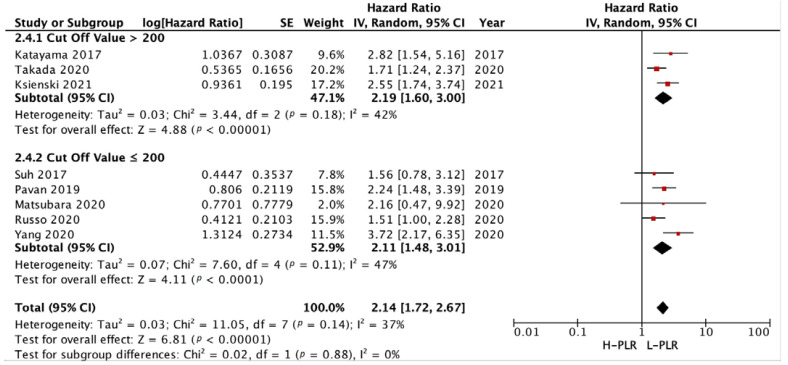
Forest plot cut-off value subgroup analysis of H-PLR versus L-PLR to OS in patients treated with immunotherapy. Red dots represent study weights; the bivalve represent the overall effect.

## Data Availability

Not applicable.

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
