# Peer review of "Neutrophil-to-Lymphocyte Ratio and Platelet-to-Lymphocyte Ratio as Prognostic Markers for Advanced Non-Small-Cell Lung Cancer Treated with Immunotherapy: A Systematic Review and Meta-Analysis"

_medicina, 2022, doi:10.3390/medicina58081069_

Round 1

Reviewer 1 Report

In the manuscript “Neutrophil-to-lymphocyte ratio and Platelet-to-lymphocyte ratio as Prognostic Markers for Advanced Non-Small Cell Lung Cancer Treated with Immunotherapy: A Systematic Review and Meta-Analysis”, Platini et al. evaluate whether the neutrophil-to-lymphocyte ratio and platelet-to-lymphocyte ratio can be prognostic markers in advanced NSCLC patients treated with immunotherapy. The authors conclude that both NLR and PLR can be considered the prognostic markers for advanced NSCLC treated with immunotherapy through reviewing current research work and meta-analysis. The topic is interesting and this work has significant clinical value.  The following points need to be considered before its  publication.

1.      Were there any effective measures taken to acquire comprehensive data that have been published or unpublished? And how to evaluate the effectiveness of the researching strategies?

2.      Publication bias and location bias should be fully considered in the meta-analysis. Funnel plots have been used to investigate the potential publication bias. However, how about the fail-safe number?

3.      NOS was used to perform the quality assessment. However,  were there any other methods also used to ensure the effectiveness of the meta-analysis, such as Thomas, Cowley and STROBE?

4.      Several reports demonstrated that NLR could independently predict the prognosis of advanced NSCLS patients, while PLR was not significantly associated with standard prognostic value. How to understand this differences in conclusion? Relevant contents should be discussed in the discussion section.

Author Response

Dear Reviewer,

Thank you for allowing us to submit a revised draft of our manuscript titled Neutrophil-to-lymphocyte ratio and Platelet-to-lymphocyte ratio as Prognostic Markers for Advanced Non-Small Cell Lung Cancer Treated with Immunotherapy: A Systematic Review and Meta-Analysis. We appreciate the time and effort you have dedicated to providing your valuable feedback on our manuscript. We are grateful to the reviewers for their insightful comments on our paper. We have been able to incorporate changes to reflect most of the suggestions provided by you. We have highlighted the changes within the manuscript. Here is a point-by-point response to the reviewers’ comments and concerns.

Comment 1: 

Were there any effective measure to acquire comprehensive data that have been published or unpublished? And how to evaluate the effectiveness of the researching strategies?

Response:

  • The underlying process that establishes the data available for analysis is an important process for systematic review and meta-analysis
  • To obtain a suitable process and comprehensive data, we used the PRISMA procedure (Fig. 1). It also comes with the appropriate keyword specifications for each database.
  • This appropriate search process is aimed at minimizing bias and providing an objective search process. 
  • For effectiveness of searching, we conduct a search through independent reviews which if differences of opinion will be reached through a common convention. A critical appraisal has also been done for each study.

Comment 2:

Publication bias and location bias should be fully considered in the meta-analysis. Funnel plots have been used to investigate the potential publication bias. However, how about the fail-safe number?

Response: 

  • Fail-safe (file-drawer) number is a method to assess the impact of publication bias; Meanwhile, funnel plot is used to identify the existence of publication bias. We do not use failsafe due to software limitations and we have acknowledged the limitations of the study

Comment 3:

NOS was used to perform the quality assessment. However, were there any other methods also used to ensure the effectiveness of the meta-analysis, such as Thomas, Cowley and STROBE?

Response:

  • To our knowledge, NOS is one of the best and most known critical appraisal in meta-analysis of observational studies (non randomized studies)
  • 'Star system' has been developed in which a study is judged on three broad perspectives: the selection of the study groups; the comparability of the groups; and the ascertainment of either the exposure or outcome of interest for case-control or cohort studies respectively
  • Furthermore, differently from other checklists and tools, it is validated for case-control and longitudinal studies (cohort).

Comment 4:

Several reports demonstrated that NLR could independently predict the prognosis of advanced NSCLC patients. While PLR was not significantly associated with standard prognostic value. How to understand these differences in conclusion? Relevant contents should be discussed in the discussion section?

Response

  • In our meta-analysis, indeed, we found that NLR was more reliable as a prognostic marker compared to PLR, proved by its hazard ratio. This result was supported by other similar meta-analysis, such as Xu, et al and Tan, et al. However, the reason / detailed mechanism of how this phenomena happens is still unknown. 
  • Subsequently, we have already included this problem as our recommendation in lines 293-295.

Warm regards,

The authors

Reviewer 2 Report

In the present systemic review authors aim to evaluate neutrophil-to lymphocyte ratio (NLR) and platelet-to-lymphocyte ratio (PLR) as prognostic markers in advanced NSCLC patients treated with immunotherapy. The work looks interesting but some issues need to be addressed. 

Major points:

1) Authors must provide more recent data of cancer statistics in the introduction section of the manuscript.

2) Authors must give more information about lung cancer subcategories (small cell lung cancer, non-small cell lung cancer etc.) in the introduction section.

3) Authors should present the methodology of the research in a more appropriate manner. Give the hypothesis-driven purpose.

4) Authors need to explain better in the results section the potential of NLR and PLR to be promising prognostic markers in immunotherapy-treated NSCLC patients.

5) If possible, authors should find and provide the N/A data presented in Tables. In table 1, I think that research No5 was performed in Italy. Could the authors clarify this?

6) Discuss, in a more detailed manner, the correlation of H-NLR and H-PLR to shorter PFS and OS.

Minor points: 

1) Line 54-55: The bibliography does not only refer to meta-analysis  of Tan.

2) Line 189: OFS ==> PFS

3) Line 195: review ==> reviews

4) Line 220-221: delete the word "lines", metastases ==> metastatic

5) Line 222 ==> delete "be"

6) Line 241 ==> MMP9?

Author Response

Dear Reviewer,

Thank you for allowing us to submit a revised draft of our manuscript titled Neutrophil-to-lymphocyte ratio and Platelet-to-lymphocyte ratio as Prognostic Markers for Advanced Non-Small Cell Lung Cancer Treated with Immunotherapy: A Systematic Review and Meta-Analysis. We appreciate the time and effort you have dedicated to providing your valuable feedback on our manuscript. We are grateful to the reviewers for their insightful comments on our paper. We have been able to incorporate changes to reflect most of the suggestions provided by you. We have highlighted the changes within the manuscript. Here is a point-by-point response to the reviewers’ comments and concerns.

Comment 1: 

Authors must provide more recent data of cancer statistics in the introduction section of the manuscript 

Response: 

Thank you, we have updated our references in lines 36-38

Comment 2:

Authors must give more information about lung cancer subcategories in introduction section 

Response:

Thank you, we have given more information about related topic in lines 38-40

Comment 3:

Authors should present the methodology of the research in a more appropriate manner. Give the hypothesis-driven purpose 

Response:

Thank you for pointing this out. We mentioned the hypothesis-driven purpose in the introduction section line 63.

Comment 4:

Authors need to explain better in the results section the potential of NLR and PLR to be promising prognostic markers in immunotherapy-treated NSCLC patients 

Response:

Thank you for your valuable input, we have added in line 272-279

Comment 5:

If possible, authors should find and provide the N/A data presented in Tables. In table 1, I think that research No 5 was performed in Italy. Could the authors clarify this? 

Response:

  • Not all included studies have the complete interest data that is needed in our meta-analysis . The “N/A” means the study does not provide data
  • Russo, et al (2018) study was done in Italy, we agreed with Reviewer 2. We have also double-checked for others.

Comment 6:

Discuss, in a more detailed manner, the correlation of H-NLR and H-PLR to shorter PFS and OS

Response: 

Thank you for valuable input, we have provided the mechanism in discussion section line 235-247. However, to date, the exact mechanism that explained poor prognostic still remains unclear. Our literature search only found NLR and PLR were associated with inflammation and tumor immune response as provided in our discussion. 

Warm regards,

The authors 

Round 2

Reviewer 1 Report

Thank you very much for your reply, and I am satisfied with the author's response. 

Reviewer 2 Report

The authors have made substantial changes and improved the quality of the manuscript rendering it appropriate for publication.

This manuscript is a resubmission of an earlier submission. The following is a list of the peer review reports and author responses from that submission.